# Knowledge, Perceptions and Attitudes of eHealth and Health Technology among Nursing Students from Gauteng Province, South Africa

**DOI:** 10.3390/healthcare11121672

**Published:** 2023-06-06

**Authors:** Eucebious Lekalakala-Mokgele, Mygirl P. Lowane, Ntlogeleng Mabina Mogale

**Affiliations:** 1School of Health Care Sciences, Sefako Makgatho Health Sciences University, Pretoria 0204, South Africa; sebi.lekalakala@smu.ac.za (E.L.-M.); mabina.mogale@smu.ac.za (N.M.M.); 2Department of Public Health, Sefako Makgatho Health Sciences University, Pretoria 0204, South Africa

**Keywords:** nursing students, eHealth, health technology, eLearning, mobile devices, digital literacy, knowledge, perception, attitude

## Abstract

The use of technology in healthcare settings was identified as a crucial priority in many countries to provide quality healthcare services. eHealth or digital health technology is found to have a positive influence on improving the efficiency and quality of healthcare services. It has been proven to provide opportunities to strengthen health systems. This study aims to assess eHealth literacy, pre-existing knowledge and the nursing students’ perceptions and attitudes towards eHealth. A quantitative, descriptive, cross-sectional survey was used in this study. The population of this study comprised 266 nursing students in the undergraduate programme of the Department of Nursing, among which 244 consented to participate in the study. A standardised and self-administered tool was utilised to collect data from nursing students across the four levels of study. Results showed that students in level four exhibit high scores of knowledge in the use of eLearning technology as compared to first-entry university nursing students. Nursing students used the internet frequently, especially to access social media and search for health and medical information for their study. Attitudes towards eHealth and technology were also found to be positive. The study recommends that digital literacy should be enhanced in the nursing education curriculum in other to further strengthen the knowledge and skills towards the use of eHealth and health technology among nursing students.

## 1. Introduction and Background

The use of technology in healthcare systems has been identified as one form of health care management transformation [1]. Electronic-Health (eHealth) or mobile-Health (mHealth) has been described as the tool for information and communication technology to reinforce healthcare management [2,3,4]. The use of technology in healthcare settings was identified as a crucial priority in many countries to provide quality healthcare services, more especially in many developed countries where they invested a lot in the e-health system [4]. It was found that eHealth has a positive influence on improving the efficiency and quality of healthcare services [4,5]. Moreover, eHealth or digital health technologies have been proven to provide opportunities to strengthen health systems [6].

The literature suggests that there is an increasing need for the implementation of eHealth [7]. Both eHealth and mHealth contribute to healthcare services by ensuring that healthcare services are provided in a cost-effective and secure manner [8]. The adoption of eHealth in healthcare services is one of the steps for opening doors towards the implementation of digital technologies [9]. Despite the potential positive benefits of eHealth [10], it was highlighted that as a result of the rapid growth of eHealth, there is a lack of consistency in defining and using terms related to eHealth. While there are concrete promises about eHealth strengthening healthcare systems, challenges exist on how eHealth will be implemented [8]. In addition, apart from the benefits of eHealth, the initiatives of eHealth in developing countries mostly failed when being evaluated [9]. The reason of failure varies, but some factors include lack of internet availability, human resource constraints and resistance to change among health professionals [9].

Although eHealth encourages communication between healthcare providers and the clients and sharing of information and knowledge between healthcare providers, the lack of a clear eHealth strategy remains the major barrier that contributes to poor healthcare quality [11]. In a study conducted in Uganda, it was identified that most of the eHealth applications and products were running in silos and were not interoperable, and it was found to prevent the sharing of information and services [11]. Furthermore, it is also highlighted that one of the contributory factors to the failure of eHealth is the fact that most healthcare cadres have low levels of computer literacy and skills to use the information and communication technologies (ICTs) required for the implementation of eHealth [12]. In order to improve the success of eHealth implementation, factors that can positively influence the outcome of the intervention should be identified [13].

Nursing students in different countries perceive the use of the internet to be useful [14]. However, there is limited literature on the use of eHealth literacy among nursing students in tertiary universities in Southern African regions. South Africa’s National Department of Health (NDoH) has taken a step forward to embrace the potential use of eHealth technologies in health sectors to improve quality of care. South Africa also has the required legislative plan in place that can be expanded to fit the requirement of growing digital health. Several initiatives have been implemented to increase access to the availability of information from healthcare settings. The initiatives have also expanded to the South African-based universities through the development of short courses and eLearning. Despite these efforts, there is still a gap in the implementation of eHealth education within nursing education curricula and the study setting, where the current strategy for undergraduate learning involves traditional lecturing, simulations and clinical practices. Therefore, there is still a gap which must be explored. Hence, this study aims to assess eHealth literacy, knowledge and the nursing students’ perception and attitude towards eHealth.

## 2. Methods

### 2.1. Study Design and Setting

A non-experimental, quantitative, descriptive, cross-sectional study was employed in this study. The study was conducted in a tertiary institution in Gauteng province.

### 2.2. Study Population

The population of this study comprised nursing students in the undergraduate programme. To avoid selection bias, the entire student population of 266 students registered for the year 2022 was invited to participate.

### 2.3. Sampling and Recruitment

From a potential target population of 266 student nurses enrolled in the nursing programme, 244 consented to participate in the study, thus yielding a response rate of 93%. Participants from study level one to four were recruited either through their course coordinators during their block classes, whereas others were recruited during their practical clinical blocks.

### 2.4. Data Collection

A self-administered questionnaire adopted from Anderberg [15] was utilised to collect data from nursing students across all four levels of study. The 41-item tool comprised seven sections which included sociodemographic details, knowledge regarding the use of technology, frequency of technology use, usage of internet, access to information on the internet, perception, and attitudes towards the use of eHealth and technology.

Informed consent was obtained from the participants prior to issuing the questionnaire. Completed forms were collated within seven days by the course coordinators and returned to the principal investigator. After data collection, the questionnaires were checked for completeness and captured using Google Forms (Google, LLC) to minimise errors. Data were automatically populated on an online Google Sheet (Google, LLC) which was then downloaded and imported into STATA 17 SE (College Station, TX, USA: StataCorp, LLC) for cleaning, coding, and analysis.

Knowledge about the use of technology such as desktop computers, laptops, tablets, and smartphones was assessed using four questions, measured on a 5-point Likert scale ranging from “Not knowledgeable at all” coded as 0, “Less knowledgeable” = 1, Somewhat knowledgeable” = 2, “Knowledgeable” = 3 and “Very knowledgeable” = 4. A summation score for each participant was derived by adding their responses across all four questions and then converted into percentages. The overall participants’ knowledge was categorised using a modified version of Bloom’s cut-off points. Scores between 80% and 100% were categorised as “Good”, scores between 50% and 79% were categorised as “Moderate” and those below 50% were categorised as “Poor” [16].

Additionally, participants answered questions about how frequently they used various devices in accessing the internet within the three months prior to the study, and they responded with either “Always”,” Sometimes”, or “Never”. The study also examined common online services along with usage frequencies, and these included services such as social media, online banking, use of electronic identification means, health applications, booking of medical appointments, blogging and health and medical information searches.

The study evaluated participants’ perceptions towards eHealth and technology using 7 questions, which were rated on a 5-point Likert scale ranging from “Strongly disagree” coded as 0 to “Strongly agree” coded as 4. The score for each question was presented as mean and standard deviation. For overall sample perception, a higher mean score (≥3) represented positive perception. Internet usage experience was assessed among participants using 8 questions which were also rated on the same 5-point Likert scale and were coded and summed up in much the same way as the knowledge questions. The modified Bloom’s cut-off points as described above were used to categorise the overall experience of the participants in using eHealth and technology. Those who scored 80% and 100% were categorised as “More experienced”, those who scored 50% and 79% were categorised as “Moderately experienced”, and those who scored less than 50% were categorised as “Less experienced”. Lastly, six questions were used to assess participants’ attitudes towards eHealth and technology, and these were also measured on the same 5-point Likert scale. The overall attitude scores were categorised using a modified version of Bloom’s cut-off points described above. Scores between 80% and 100% were categorised as “More favourable”, those between 50% and 79% were categorised as “Moderately favourable” and scores below 50% were categorised as “Less favourable”. For both participants’ experience and attitude on eHealth and technology, overall mean scores were computed and presented as mean and standard deviation. Similarly higher mean scores (≥3) denote that the participants were more experienced and had a more favourable attitudes towards eHealth and technology, in general.

### 2.5. Data Analysis

Descriptive statistics were used to analyse data. Frequency distribution was utilised to analyse all categorical variables to determine relative frequencies. All numerical variables were analysed using summary statistics, and these were presented as means and standard deviations (SD). The findings of the study are presented in tables and graphs. Pearson’s chi-square test was used to compare categorical variables. The skewness and kurtosis normality test was used to assess the distribution of the knowledge scores. The scores were not normally distributed, and as such, the Kruskal–Wallis one-way analysis of variance (ANOVA) was used to compare the knowledge scores across the different study levels of study. For pairwise group comparison of the knowledge scores, Dunn’s post-estimation test with Bonferroni correction was used. Statistical significance was set at *p*-value < 0.05.

### 2.6. Reliability, Validity and Bias

Cronbach’s alpha values for questions measuring knowledge, perceptions, attitudes and experiences was calculated to ensure internal consistency reliability, which were 0.71, 0.79, 0.87 and 0.90, respectively. The tool was reviewed by two panel experts for content validity. To minimise selection bias, all 266 registered students across the four levels of study were invited to participate. Recall bias was minimised by asking participants to recall events within the last three months prior to data collection. To minimise social desirability bias and encourage honest responses from the participants, the questionnaire was anonymous.

### 2.7. Ethical Consideration

Permission to conduct research was sought from the Institutional Research Ethics Committee, and written informed consent was obtained from the participants prior to collecting data. All data were delinked at the outset by separating the consent forms and completed questionnaires to ensure anonymity.

## 3. Results

### 3.1. Sociodemographic Characteristics

Table 1 below shows the sociodemographic characteristics of the participants in the study. Our results show that the mean age of the participants was 22 years (sd = 2.97) with a minimum age of 17 years and a maximum age of 34 years. Most participants (64%) were in the 20–24 years age group, which was followed by those who were less than 20 years. There were more female students than males (73% vs. 27%). In terms of level of study, more participants (36%) were in their final year of study.

### 3.2. Knowledge Regarding the Use of Technology

Seventy-seven percent (77%) of the participants indicated they were knowledgeable about using a desktop computer as compared to 23% who did not, (see Table 2). The majority of students (88.51%) who knew how to use a desktop computer were in their fourth level of study, which was followed by students in their third, second and first level of study. There was a statistically significant difference in the proportion of participants who were knowledgeable in using desktop computers among the students (*p* = 0.006). A similar trend was observed with the knowledge of using a laptop computer with the majority (97.70%) being in the fourth level of study. In terms of the knowledge of using a tablet, first-year students were the second highest after the fourth-year students. Interestingly, all first-year students were able to use smartphones compared to some third-year students (91.89%).

Overall, 71.90% (174/242) of participants had good knowledge on the use of technology, which was followed by 25.21% who had moderate knowledge. Our data further show an overall mean knowledge score of 89% (SD 19.97) among the study population. The results from the Kruskal–Wallis ANOVA showed that there is a statistically significant difference between the overall scores across the different levels of study (*p* = 0.006). The pairwise comparison using Dunn’s test indicated that the knowledge scores for level-four nursing students were significantly different from knowledge scores of level-one nursing students (4 vs. 1, *p*-value = 0.002). No significant differences were observed between other groups.

### 3.3. Frequency of Technology Used to Access the Internet in the Past 3 Months

Figure 1 below shows the frequency of use of technology over the past 3 months among the study participants. The results indicate that most students (95%) always used a smartphone (daily or several times in a day) to access the internet, which was followed by a laptop computer (85%), which is also used daily or several times per day. The least used technologies to access the internet were a tablet (51%) and a desktop computer (25%). Although the results are not shown, no statistically significant difference was observed regarding the frequency of desktop use (*p* = 0.386), laptop (*p* = 0.659), tablet (*p* = 0.265) and smartphone use (*p* = 0.705) across the four levels of study, respectively.

### 3.4. Most Common Uses of the Internet among Participants

The results in Figure 2 show that most students (79%) always use the internet to access social media, which was followed by searching for health and medical information (57%) and health or exercise applications (30%). The least frequently utilised function on the internet was for the use of electronic self-identification (<30%), followed by blogging (<20%), and the most least utilised was for booking medical appointments (<10%). No statistically significant differences were observed for each of the uses across the four levels of study.

### 3.5. Perception towards the Use of eHealth and Technology

Table 3 below shows the perception of nursing students towards the use of technology. The results indicate that most participants (those who agreed (*n* = 71, 29.10%) and strongly agreed (*n* = 152, 62.30%)) felt that using technology makes their lives easier (91.4%), is fun (83%) and that people who do not have access to internet are disadvantaged and missing out (77%). Furthermore, the majority of the participants indicated that they found it easy to use technology (75%), they like to acquire the latest technology models or updates (74%) and would have liked to try new technological gadgets given the support (73%). Slightly more than 50% of the participants felt that too much technology makes society vulnerable. The total mean score for the sample was 3.07 (standard deviation: 0.65), representing positive perception. No statistically significant difference in perception was observed across the four levels of study (*p* = 0.397).

#### Experience Regarding Internet Usage among Participants

The results on Table 4 below show that almost 70% of the participants knew what health resources are available on the internet, knew where to find helpful health resources (75%), knew how to find helpful health resources (75%) and knew how to use the internet to answer questions about health (80%). Furthermore, about 80% knew how to use the information obtained from the internet to assist themselves. In terms of the required skills to evaluate the health resources obtained from the internet, 64% indicated as such. Less than 50% of the participants were confident in their ability to discriminate between high- and low-quality health resources, and 54% felt confident in using the information from the internet to make health decisions. Overall, the majority of the participants (50%) were moderately experienced in using the internet with an overall mean score of 2.84 (0.72). No statistically significant difference was observed across all four levels of study (*p* = 0.314).

### 3.6. Attitudes towards eHealth and Technology

The results regarding the attitude of nursing students towards eHealth and technology are displayed in Table 5. The results show that most students (77.5%) believed that different forms of technical knowledge will be an important competence in their future work as nurses. About 59% thought that they have gained enough knowledge about eHealth in their nursing education to feel secure in their future professional roles. Almost 80% indicated they would like more eHealth in the operational sections of their theoretical courses and 82% indicated they would like more eHealth in their education in general. Seventy-eight percent of the participants believed that the knowledge of eHealth will be necessary for them to carry out good nursing to become competent nurses. Lastly, 78% thought there were many other areas that are more important for a nurse to gain more knowledge about eHealth. Overall, most participants (61.48%) had a more favourable attitude towards eHealth and technology with an overall mean score of 3.03 (0.73). No statistically significant difference was observed across the four levels of study (*p* = 0.355).

## 4. Discussion

Our study assessed eHealth literacy, pre-existing knowledge and the nursing students’ perception and attitude towards eHealth. The eHealth literacy encompasses a set of skills and knowledge that is important for the valuable engagement with technology-based health tools [17]. The eHealth literacy is an important student learning outcome that requires careful consideration by nursing educators to ensure that nursing students develop skills beyond internet searching and reading internet information and also can sort reputable and credible information from the disreputable and unsubstantiated [18,19]. Successful online learning depends on the supported learner’s attitude, motivation, and self-efficacy along with their ability to use the flexibility and convenience of digital technology. Since the internet became a major source of health-related information, nurses should have acceptable eHealth literacy [14].

This study assessed the knowledge of the use of electronic devices among the nursing students, and it was found that nursing students had excellent knowledge of the use of technology. In contrast, a study conducted in Nepal revealed that despite increased awareness of the usefulness and importance of the internet in obtaining health information, nursing students still have a moderate perceived level of eHealth literacy [14]. Furthermore, the nursing students in that study reported that they were unsure of how to use the internet and where to find useful health resources, citing the reason that they were not taught about the importance of these knowledge and skills [14]. Nursing students are expected to use technology devices for eLearning. Hence, it was found in our study that the smartphone was the most favourable as compared to laptops and other devices such as desktops. Smartphones are mobile devices that can be used as diagnostic and therapeutic tools that are as irreplaceable such as for example the stethoscope, which has been used in traditional nursing practice [20,21]. Mobile devices are small, portable, and powerful enough to do many of the same things as a desktop or laptop computer [22]. Mobile devices provide access to any information, functions, and mobile learning that can happen independently anywhere, anytime, and in real life [23]. Nursing students (NSs) can access necessary information more quickly, make notes, and make their studies more effective as well as increase their learning space [24].

It was found in this study that students in level four exhibit high scores of knowledge on the use of eLearning technology as compared to first-entry university nursing students. We assume that nursing students who do not have an underlying background in the use of eLearning technology relating to mobile devices might have challenges in adapting to the new way of learning. The literature cautions that using mobile learning might not meet the institutional expectation as anticipated; however, this does not prohibit most universities from working hard at enhancing and adopting learning processes and to provide possible support to both academic staff members and students [25].

The literature highlights that the use of the internet and social media platforms to access eHealth issues has gained popularity [26]. Our study found that most nursing students used the internet frequently, especially to access social media and search for health and medical information for their studies. Studies warn that the use of personal laptops, mobile devices, the internet, and social media can be distracting for the teaching and learning process if not monitored [27,28,29]. Despite the challenges in the use of technology for teaching and learning [30], the study has emphasised there is no way to shy away from new developments, because the use of mobile devices for teaching and learning has become prevalent due to the rapid development and updated technology [31]. Adding to that, social media was proven to provide pedagogical, social, and technological benefits and has the potential to be diversely utilised in higher learning institutions [32,33].

Perceptions and attitudes towards eHealth and health technology were assessed in this study and found to be highly positive in almost all the participants. Similar findings were also found in a study conducted in the United States of America (USA) among nurses where acceptability was characterised by increased perceptions (perceived usefulness) and positive feelings along with increased intentions to use their mobile devices [34,35]. Another study also supports positive sentiments and attests this to the fact that students belong to a generation raised with information, which makes them more confident in manipulating new platforms and devices and with little or no training in the area [36]. The participants shared their sentiments about using technology for their teaching and learning and reported it to be interesting. Thanks to the existing enthusiasm demonstrated by the study participants, we are confident that the use of technology will be on the rise. One study conducted on digital educational technologies perspective among nursing students found that the use of this technology was a major benefit, enhances motivation for learning, and improves learning experiences, particularly in clinical learning settings [37]. However, the use of digital health and eHealth platforms in nursing education is not fully exploited [38], and there is very limited research investigating students’ understanding of eHealth [39], so it is crucial to prepare and support future generational nursing students to use digital health and eHealth [40]. Although using digital health and eHealth entails challenges that nursing educators and students must deal with to provide effective learning environments [38], nursing educators must ensure that digital health and eHealth must be integrated into the nursing curriculum and teaching environment [40].

The findings in this study imply that there is a need to develop a plan for continuous education that will be aligned with the requirement of the end-user in practice [2]. McCabe [3] added that education, theory, and practices should be integrated at the undergraduate level and throughout the career. Although our study was not aimed to inform eHealth teaching at our institution, nursing educators should empower newly enrolled student nurses with knowledge, skills, motivation, and confidence to prepare them for the future use of nursing and health technologies. Thus, a lack of information and communication technologies skills and knowledge among student nurses might contribute to emotional resistance and fear of using digital technology. The findings of Lam [39] suggest that when identifying eHealth learning needs, it enhances learning and teaching practices and thereby increases students’ engagement.

## 5. Limitation

The participants were sampled through a convenience sample consisting of one cohort of nursing students. Therefore, the study cannot be generalised to all undergraduate students across teaching facilities. In addition, previous experience with the use of digital technologies before enrolling in tertiary education was not assessed during the study. Data analysis was not disaggregated by gender. This study did not assess whether the use of mobile devices offers benefits or barriers in teaching and learning and students’ progress, therefore considering it as a limitation.

## 6. Conclusions and Recommendations

The result shows that a significant number of nursing students were aware of the need to use eHealth and health technology, as evidenced by their frequency of use. Smartphones and laptop computers, for access to the internet, were found to be most utilised. Although many current nursing students seem to be well skilled in eHealth and technology, the study found variation in knowledge regarding the use of mobile devices, the internet, laptops, tablet devices, smartphones, and desktops among the nursing students. The nursing students in years three and four were the ones with better knowledge than the first-entry students. The study recommends that first-entry students who were previously disadvantaged or poorly prepared in high schools should also be supported to overcome barriers that might affect the success of their learning in the university environment.

Moreover, the study also found that almost all nursing students who participated in this survey have a positive attitude towards the use of eHealth and health technology. As the use of information technology increases, the necessary skills and positive attitude for digital literacy should be an essential aspect for the further enhancement of eHealth and health technology in nursing education. Given the fact that there might still be gaps in the understanding of eHealth, it will be imperative to further investigations regarding the preparedness of an eHealth-enabled environment. Our study, therefore, recommends that future research can be conducted to investigate and offer recommendations for the development of a nursing education curriculum that will improve digital literacy towards eHealth learning among nursing students.

## Figures and Tables

**Figure 1 healthcare-11-01672-f001:**
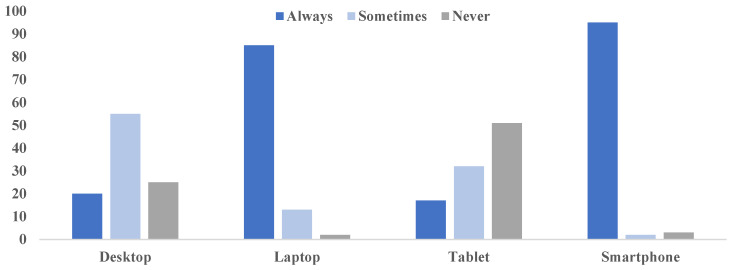
Frequency of technology used to access internet in the past 3 months.

**Figure 2 healthcare-11-01672-f002:**
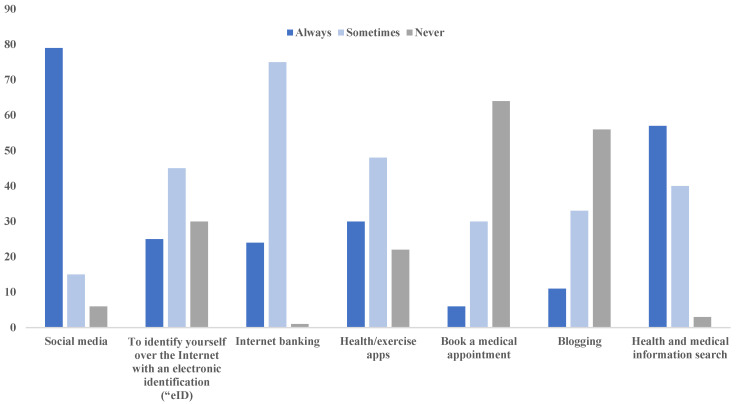
Most common internet uses among participants.

**Table 1 healthcare-11-01672-t001:** Sociodemographic characteristics of nursing students.

Variables	*n*	%
Age		
<20 years	57	23.36
20–24 years	155	63.52
25–30 years	30	12.3
>30 years	2	0.82
Sex		
Female	177	72.54
Male	67	27.46
* Level of study		
1	61	25.21
2	57	23.55
3	37	15.29
4	87	35.95

* *n* < 244 due to non-response to level of study.

**Table 2 healthcare-11-01672-t002:** Knowledge regarding the use of technology.

							Kruskal–Wallis ANOVA	
	Level 1	Level 2	Level 3	Level 4	Total	*p*-Value	Groups	*p*-Value
	*n* (%)	*n* (%)	*n* (%)	*n* (%)	*n* (%)			
**Desktop**								
Yes	39 (63.93)	43 (75.44)	28 (75.68)	77 (88.51)	187 (77.27)	0.006		
No	22 (36.07)	14 (24.56)	9 (24.32)	10 (11.49)	55 (22.73)			
**Laptop**								
Yes	52 (85.25)	53 (92.98)	35 (94.59)	85 (97.70)	225 (92.98)	0.034		
No	9 (14.75)	4 (7.02)	2 (5.41)	2 (2.30)	17 (7.02)			
**Tablet**								
Yes	55 (90.16)	47 (82.46)	32 (86.49)	83 (95.40)	217 (89.67)	0.082		
No	6 (9.84)	10 (17.54)	5 (13.51)	4 (4.60)	25 (10.33)			
**Smartphone**								
Yes	61 (100)	56 (98.25)	34 (91.89)	86 (98.85)	237 (97.93)	0.039		
No	0 (0.00)	1 (1.75)	3 (8.11)	1 (1.15)	5 (2.07)			
Overall knowledge								
Poor knowledge *n* (cell%)	2 (0.83)	2 (0.83)	3 (1.24)	0 (0.00)	7 (2.89)	0.012		
Moderate knowledge *n* (cell%)	23 (9.50)	16 (6.61)	8 (3.31)	14 (5.79)	61 (25.21)			
Good knowledge *n* (cell%)	36 (14.88)	39 (16.12)	26 (10.74)	73 (30.17)	174 (71.90)			
Knowledge score% mean (SD)	85 (21.05)	87 (22.22)	87 (26.10)	95 (11.95)	89 (19.97)	0.006	4 vs. 1	0.002

**Table 3 healthcare-11-01672-t003:** Perception towards the use of eHealth and technology.

	Strongly Disagree	Disagree	Neutral	Agree	Strongly Agree	Mean Score
	*n* (%)	*n* (%)	*n* (%)	*n* (%)	*n* (%)	Mean (SD)
1. Using technology makes life easier for me	7 (2.87)	2 (0.82)	12 (4.92)	71 (29.10)	152 (62.30)	3.47 (0.86)
2. I think it is fun with new technological gadgets	8 (3.28)	2 (0.82)	32 (13.11)	78 (31.97)	124 (50.82)	3.26 (0.95)
3. People who do not have access to the internet have a real disadvantage because of all what they are missing out on	12 (4.92)	11 (4.51)	33 (13.52)	60 (24.59)	128 (52.46)	3.15 (1.12)
4. I find it easy to learn how use new technology	10 (4.10)	6 (2.46)	44 (18.03)	105 (43.03)	79 (32.38)	2.97 (0.98)
5. I like to acquire the latest models or updates in technology	9 (3.69)	12 (4.92)	41 (16.80)	84 (34.43)	98 (40.16)	3.02 (1.05)
6. I would have dared to try new technological gadgets to a greater extent if I had more support and help than I have today	9 (3.69)	9 (3.69)	46 (18.85)	85 (34.84)	95 (38.93)	3.01 (1.03)
7. Too much technology makes society vulnerable	13 (5.33)	35 (14.34)	63 (25.82)	65 (26.64)	68 (27.87)	2.57 (1.18)
Overall mean perception score						3.07 (0.65)

**Table 4 healthcare-11-01672-t004:** Experience regarding internet use among participants.

	Strongly Disagree	Disagree	Neutral	Agree	Strongly Agree	Mean Score
	*n* (%)	*n* (%)	*n* (%)	*n* (%)	*n* (%)	Mean (SD)
1. I know what health resources are available on the internet	6 (2.46)	3 (1.23)	65 (26.64)	106 (43.44)	64 (26.23)	2.89 (0.89)
2. I know where to find helpful resources on the internet	5 (2.05)	8 (3.28)	49 (20.08)	104 (42.62)	78 (31.97)	2.99 (0.92)
3. I know how to find helpful health resources on the internet	5 (2.05)	7 (2.87)	49 (20.08)	108 (44.26)	75 (30.74)	2.99 (0.81)
4. I know how to use the internet to answer my questions about health	7 (2.87)	7 (2.87)	33 (13.52)	115 (47.13)	82 (33.61)	3.05 (0.92)
5. I know how to use the health information I find on the internet to help me	8 (3.28)	9 (3.69)	32 (13.11)	115 (47.13)	80 (32.79)	3.02 (0.95)
6. I have the skills I need to evaluate the health resources I find on the internet	6 (2.46)	11 (4.51)	71 (29.10)	107 (43.85)	49 (20.08)	2.75 (0.91)
7. I can tell high-quality health resources from low-quality health resources on the internet	9 (3.69)	24 (9.84)	91 (37.30)	78 (31.97)	42 (17.21)	2.49 (1.00)
8. I feel confident in using information from the internet to make health decisions	12 (4.92)	26 (10.66)	75 (30.74)	84 (34.43)	47 (19.26)	2.52 (1.07)
Overall mean experience score						2.84 (0.72)
Overall experience of using the internet: Less experienced 6.65% *(n* = 16); Moderately experienced 50% *(n* = 122) and More experienced 43.44% *(n* = 106)						

**Table 5 healthcare-11-01672-t005:** Attitudes towards eHealth and technology.

	Strongly Disagree	Disagree	Neutral	Agree	Strongly Agree	Mean Score
	*n* (%)	*n* (%)	*n* (%)	*n* (%)	*n* (%)	Mean (SD)
1. I believe that different forms of technical knowledge will be an important competence in my future work as a nurse	9 (3.69)	4 (1.64)	42 (17.21)	82 (33.61)	107 (43.85)	3.12 (0.99)
2. I think that so far, I have gained enough knowledge about eHealth in my nursing education to feel secure in my future professional role	12 (4.92)	20 (8.20)	69 (28.28)	92 (37.70)	51 (20.90)	2.61 (1.05)
3. I would like more eHealth in the operational parts of my theoretical courses	5 (2.05)	6 (2.46)	39 (20.49)	111 (45.49)	83 (34.02)	3.06 (0.88)
4. I would like more eHealth in the operational parts of my education	6 (2.46)	6 (2.46)	32 (13.11)	108 (44.26)	92 (37.70)	3.12 (0.90)
5. I believe knowledge of eHealth will be necessary to carry out good nursing to be a competent nurse	3 (1.23)	8 (3.28)	43 (17.62)	93 (38.11)	97 (39.75)	3.11 (0.89)
6. I think there are many other areas that are more important for a nurse to gain more knowledge about the eHealth	5 (2.05)	4 (1.64)	44 (18.03)	89 (36.48)	102 (41.80)	3.14 (0.91)
Overall mean attitude score						3.03 (0.73)
Overall attitude towards eHealth and technology: Less favourable 4.51% *(n* = 11), Moderately favourable 34.02% (*n* = 82) and More favourable 61.48% (*n* = 150).						

## Data Availability

All the data are included in the analysis of the study and shall be available upon reasonable request.

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
