# Peer review of "Knowledge, Perceptions and Attitudes of eHealth and Health Technology among Nursing Students from Gauteng Province, South Africa"

_healthcare, 2023, doi:10.3390/healthcare11121672_

Round 1

Reviewer 1 Report

This is an interesting study that assesses the knowledge, perception and attitude of nursing students towards eHealth at a university in South Africa.
The objectives of the study are of interest to the scientific community, but the article has a number of limitations that, in my opinion, should be corrected before publication.
Among the main limitations of the paper are the following:
- The abstract should include the objective of the study
- The questionnaire used for data collection should be adequately described.
- The statistical methods used should be clarified
- The data analysis performed is not well explained.
- The results should be analysed by gender as there are often significant differences in technological issues.
- The discussion should reflect on the reasons that might have favoured the results of the study and then compare the results with similar studies.
I have pointed out these and other aspects that should be corrected in the contributions that I have included, in the form of comments, in the attached document.
Kind regards

Author Response

Thank you for giving us the opportunity to revise our manuscript. Hope we have done justice to the comments. the comments were informative, learning curve and eye opening for future studies.

Reviewer 2 Report

Data Collection section begins stating a standardized questionnaire was used.  On further reading, to me, it appears that several sets of questions that have been validated individually are included (such as eHEALS by Norman and Skinner) but those details and references to the originators are lacking.  I think each of the question sets should be described. Is there a standardized questionnaire available with all the questions used by the authors? The transcription from the original questionnaire to Google Forms is confusing, or did the researchers use Google Forms to collect data from the students? The paragraph starting on line 104 that talks about the recoding of variables is confusing and can use further detail, section by section, with rationale of why the collapsing was necessary?  Were these determinations per the literature review suggestions, etc?  In some questions, it would seem that the answers would need to be transposed such as "Too  much technology makes society vulnerable" which would be in a different direction to the other questions in that set (Perceptions towards eHealth and health technology).

Is Table 1 broken into two tables, one for demographics and one for Knowledge regarding use of technology? Lines 184-186, this needs to be reworded, it seems like it pertains to the prior sentence and the % don't make sense. The rest of the tables, 3-5, are presented per the questionnaire, but the narrative is per the method of analysis which combined/collapsed answer choices into two categories. Would it be better to show data in the tables that matches the analysis, with perhaps full results in an appendix?

In the Discussion, wouldn't you also assume students in the fourth year of study have done internships requiring use of technology to improve their knowledge, skills and confidence?

Overall, I think this was an interesting study of an important population in health care.  Its possible that two papers could come out of the findings, as it feels like a lot is trying to be accomplished here but details are lacking.

Overall ok, line 14, rather than "however only" I suggest with 244 consenting to participate.  You have a very high response rate, and "only" should be removed from line 86 as well. Line 38, "opening governance"? Not sure what that means, need further explanation. Otherwise writing is fine.

Author Response

Thank you for giving us the opportunity to correct our manuscripts. we have learnt a lot from your comments. Hope we have done justice without changing the initial submission. 

Round 2

Reviewer 1 Report

The authors have done an important job of revision, incorporating the contributions made in an appropriate manner.
As aspects that could still be improved, I would point out the following:
- Include in the limitations the fact that the analysis was not disaggregated by gender (which is very relevant in this issue of technology use).
- In Table 1 and in the presentation of these results (Knowledge regarding the use of technology) the percentages should refer to each year. That is, of the first year students, what percentage is knowledgeable and what percentage is not. And not, as is currently the case, showing the % of students who have knowledge, since this figure is highly conditioned by the greater participation of 4th year students.
- In the results for 'Frequency of use of technology' and 'Internet use', they should describe whether there are differences by year.
- In section 5.2 you could describe how you analysed the normality of the data distribution.
- You could include the questionnaire used in an annex.

Kind regards

Author Response

Dear Reviewer 

Thank you once more for the opportunity provided to revise our manuscript. All the necessary inputs implemented and highlighted in yellow in the main document.

Reviewer 2 Report

Two areas I will try to clarify:  1. Data Collection section begins stating a standardized questionnaire was used. On further reading, to me, it appears that several sets of questions that have been validated individually are included (such as eHEALS by Norman and Skinner) but those details and references to the originators are lacking. I think each of the question sets should be described. = we are not clear with is comment. However, 2.4 data analysis section has been improved. See line 103 -130 

NEW COMMENT: My question is the origins of the questions you used in the questionnaire.  I know some of them are from the eHEALS instrument developed by Norman and Skinner (Table 4) and this should be noted in the Methods section.  Where did the other question sets originate?  Each originator should be given credit.  Has a prior researcher/author combined the question sets together as you have (in other words validated the full question set)?

7. The rest of the tables, 3-5, are presented per the questionnaire, but the narrative is per the method of analysis which combined/collapsed answer choices into two categories. Would it be better to show data in the tables that matches the analysis, with perhaps full results in an appendix? =Comment not clear. However the variables in these tables were the once in the questionnaire. The analysis of likert scale questions were guided by the following article: Taherdoost H. What is the best response scale for survey and questionnaire design; review of different lengths of rating scale/attitude scale/Likert scale. Hamed Taherdoost. 2019 Mar 29:1-0.

NEW COMMENT: The authors describe how Likert answers were collapsed from 5 to 2 for analysis on the original version. I was looking for a rationale as to why this was done, is that what Taherdoost describes in the above article? If so, perhaps adding that rationale will be helpful.

Author Response

(The authors gave the same response as above.)
